# Wearable IoT Smart-Log Patch: An Edge Computing-Based Bayesian Deep Learning Network System for Multi Access Physical Monitoring System

**DOI:** 10.3390/s19133030

**Published:** 2019-07-09

**Authors:** Gunasekaran Manogaran, P. Mohamed Shakeel, H. Fouad, Yunyoung Nam, S. Baskar, Naveen Chilamkurti, Revathi Sundarasekar

**Affiliations:** 1University of California Davis, Davis, CA 95616, USA; 2Faculty of Information and Communication Technology, Universiti Teknikal Malaysia Melaka, Durian Tunggal 76100, Melaka, Malaysia; 3Biomedical Engineering Department, Faculty of Engineering, Helwan University, Helwan 11792, Egypt; 4Department of Computer Science and Engineering, Soonchunhyang University, Asan 31538, Korea; 5Department of Electronics and Communication Engineering, Karpagam Academy of Higher Education, Coimbatore, Tamil Nadu 641021, India; 6Department of Computer Science and IT, La Trobe University, Melbourne 3086, Australia; 7Anna University, Tamil Nadu 600025, India

**Keywords:** multi access physical monitoring system, multimedia technology, edge computing, Bayesian neural network, smart-log patch

## Abstract

According to the survey on various health centres, smart log-based multi access physical monitoring system determines the health conditions of humans and their associated problems present in their lifestyle. At present, deficiency in significant nutrients leads to deterioration of organs, which creates various health problems, particularly for infants, children, and adults. Due to the importance of a multi access physical monitoring system, children and adolescents’ physical activities should be continuously monitored for eliminating difficulties in their life using a smart environment system. Nowadays, in real-time necessity on multi access physical monitoring systems, information requirements and the effective diagnosis of health condition is the challenging task in practice. In this research, wearable smart-log patch with Internet of Things (IoT) sensors has been designed and developed with multimedia technology. Further, the data computation in that smart-log patch has been analysed using edge computing on Bayesian deep learning network (EC-BDLN), which helps to infer and identify various physical data collected from the humans in an accurate manner to monitor their physical activities. Then, the efficiency of this wearable IoT system with multimedia technology is evaluated using experimental results and discussed in terms of accuracy, efficiency, mean residual error, delay, and less energy consumption. This state-of-the-art smart-log patch is considered as one of evolutionary research in health checking of multi access physical monitoring systems with multimedia technology.

## 1. Introduction

In the recent past, several studies have highlighted the importance of multi access physical monitoring systems for observing physical activities of human, which help to infer and analyse the healthcare treatment process of the human body [1]. Through the formation and application of the highly accurate interconnected health monitoring system, we can predict the health conditions of humans using rapid healthcare services in psychiatric emergency service (PES) [2]. In the near future, the medical industry will integrate multimedia technology, artificial intelligence, intelligent Internet of Things, and other high-end technologies to promote the enhancement of wearable Internet of Things (IoT) medical devices. Among all the technologies, multimedia technology is considered as one of the most prominent and cost-effective technologies, which provides quality heterogeneous healthcare data to the support healthcare of the patients. In the revolution of multimedia technology [3,4], smart patches or chips have just begun, which may be the next skin-like wearables that are capable of monitoring human health conditions using IoT sensors. These patches have an ultra-thin flexible patch array, as shown in the Figure 1, which can monitor skin temperature and function of heart; moreover, these flexible patches contain thermal liquids and imaging sensors, which help to monitor the temperature variation for human body condition [5].

This device monitors the psychological signals of the human body and provides early warning to the medical practitioner in case the health is potentially in a dangerous condition. This device consists of Bluetooth silicon on chip (SoC) for dataset transfer, battery power backup holder, and test points for data processing.

The existing patches [6] in the smart healthcare sector of the multi access physical monitoring system with multimedia technology are operated in a cloud environment to monitor the physical activities. The cloud computing technology (CCT) [7] helps to transfer the health data, which are collected and processed by IoT devices through internet using various deep learning, machine learning, and convolutional neural networks, which are deployed in the cloud environment. However, there is a massive explosion that has taken place in the present era of the multi access physical monitoring systems, where the major concerns in competent condition during data processing faced by the researchers in the era are listed below:
**Streaming of large datasets:** Continuous generation of large datasets that are deployed and distributed in cloud platforms results in congestion due to a larger amount of data processing [8]. Dataset on heterogeneity suffers more residual error [9].Improper space time relation on IoT devices in data processing leads to high noise data, which suffers errors, inaccurate data transmission in multi access physical monitoring systems, and network congestion with more cost and more energy [10].

These features increase the complexity of the wearable IoT devices, and it fluctuates the optimization parameters such as latency, throughput, accuracy, efficiency, mean residual error, delay, and more energy consumption. This unpredictable response of the device between cloud and IoT leads to network congestion with various I/O challenges in delivering reliable healthcare data for multi access physical monitoring system. The response times of the present system with IoT sensors are much less due to interrupted and discontinuous data transmission with long-range data processing intervals [11]. In addition, the privacy of the healthcare data is completely accessed by the third party, which leads to authentication problems. This leads researchers to pay their attention to designing and developing a novel system with enhanced algorithmic computation. Hence, current multi access physical monitoring systems need highly intelligent delay sensitivity with reliable health monitoring system to provide accurate data during the diagnosis of human health. In this research, a wearable smart-log patch with Internet of Things (IoT) sensors has been designed and developed with multimedia technology to overcome the incompetency, and edge computing on Bayesian deep learning network (EC-BDLN) algorithm has been used to infer and identify various physical data collected from humans in an accurate manner. This system would be a more robust and promising way to solve the problems that are presently faced in the healthcare sector of multi access physical monitoring systems in human physical activities as well as health monitoring.

### Contribution

A novel optimized neutral network with densely connected layer for determining the temperature imbalance in health;Bayesian deep learning network for accurate prediction of improper working of organs, which are integrated in Wearable IoT smart patch for data processing;Complete physical monitoring system using multimedia technology with edge computing using agile learning for real-time data analysis using IoT sensors;Streamlined efficient model to identify the various signal patterns of the human physical activities using edge computing on Bayesian neural network.

The remaining sections of the article are organized as follows. Section 2 surveys various literatures about the significance of multi access physical monitoring system and the importance of multimedia in healthcare data analysis. Section 3 describes the architecture of smart-log patch with Internet of Things (IoT) sensors along with edge computing on Bayesian deep learning network (EC-BDLN) algorithm for data computation. In Section 4, experimental validation and its discussion in comparison with existing techniques are analysed. Section 5 concludes the research with future extension.

## 2. Related Works

Presently, edge computing [12] and fog computing [13] are considered as among the most capable technologies for analysing the data sources for various application scenarios in the healthcare sector. In addition, mobile edge computing is the emerging technology, which has been in current practice of multi access physical monitoring systems. Though these technologies are rated high with promising results, the major setback for the above-mentioned technologies is latency and accuracy in transferring health datasets over the network. The usage of neural network-based mathematical computation on health dataset processing is unable to achieve efficient results in terms of reliability and energy consumption [14]. There has been significant clinical research in the field of multi access physical monitoring systems, which are mainly focused on reducing the health risk of humans. Few studies suggested that telehealth systems are practiced everywhere, which does not lead to good outcomes, however data-driven approaches are used to detect the multimodal changes of the physiology that are in practice at present [15]. This method can achieve 80% predictive ratio, with more accuracy and complexity making it unsuitable for human health analysis [16]. A group of researchers from Stanford University conducted a survey to analyse the physical activities and heart patterns of the humans using recurrent neural algorithm (RNA) and convolution neural algorithm (CNA), which achieved good results [17,18]. Data intensive analysis (DIA) has been used in the recent past, which provides location-aware sensitive monitoring of human health with more energy consumption and higher error rate. In the recent years [19], conditional-based monitoring (CBM) has been used to monitor the various signals of our human body in the multi access physical monitoring systems for the early detection of faults and other irregular activities of the organs [20]. Here, due to mixing of unwanted frequency, it provides little or is unable to provide proper information about the human health [21]. The dynamic behaviour of this algorithm makes it unsuitable for diagnosis [22]. The progress of deep learning network [23] and high prediction in classifying health datasets [24] or physical activity information [25] is deployed as one of the promising choices to solve the multi access physical monitoring system problem [26]. In [27], the IoT-based system has been introduced and it is integrated with wearable sensor nodes to analyse the human pain using facial surface electromyogram, wherein the efficiency of the device is less compared to the proposed technique discussed in this article, and in [28], the energy consumption problem is addressed through smart plug and play system using IoT. In this article, the authors fail to concern the trade-off between error and delay.

## 3. Edge Computing on Bayesian Deep Learning Network for Physical Education System Using Multimedia Technology

As shown in Figure 2, edge computing uses a hybridized platform of edge and cloud to solve the storage problem of large amounts of physical datasets. The health-related datasets are processed at edge computing platform, which consists of an IoT sensor physical monitoring device layer, edge layer, and smart log system with smart patch for processing of IoT data with multimedia technology from the human physical system. At the IoT physical monitor in, several bio sensors such as blood, temperature, electro-myo-gram (EMG), electro-cardio-gram (ECG), electro-encephalo-gram (EEG), pressure, visual, respiration, accelerator gyroscope, and sink node have been integrated with the edge platform for accurate diagnosis and prediction of body patterns. Here, edge computing technology brings data more closely to the location where the data are needed using distributed device. This wearable smart log patch with IoT sensor in edge computing environment helps to produce accurate data about the physical activities of the human physical system, which would be more useful in multi access physical monitoring systems for health monitoring of children and adults. In the past, there have been two different ways to analyse the monitored data: Expert diagnosis;Cloud storage for online diagnosis.

In both the categories, early warning about the improper function of organs is more complex and it takes more time for report generation. This has been overcome through the edge platform because it runs distributed networks using a smart router, storage unit, and high-power capacities, which are more suitable for multi access physical monitoring systems for physical monitoring of the human body. In this proposed architecture, a wearable smart patch with IoT sensors transmits the datasets to the edge platform using the local area network such as Wi-Fi and Bluetooth. In this approach, a Bayesian deep learning network algorithm, which has been used in distributed device of the edge computing environment that helps to infer and identify various physical data collected from the humans in an accurate manner to monitor their physical activities as shown in Figure 3. The initial process of this network is to analyse and extract the features or patterns of the health datasets. The normalized dataset has been processed to minimize the data reliability and redundancy. Here, the system consists of an input layer, multilayer or hidden layer, and output layer. These layers are integrated with the Bayesian network where the matrix consists of various IoT datasets as shown in Table 1, which are represented as signals and these signals are converted as numerical in the matrix; each layer is connected between the same and cross layers, which are represented as a regression model. This resembles the flow of the human brain with a limited subset. Here, the normalization operation has been mathematically derived using “Mean (µ) and Standard Deviation (*σ*)”.


**Case: 1-Mean µ = ”0”and Standard Deviation (σ) = ”1”**



**Solution:**


Inside the smart log patch of the wearable device, the input of the network is represented as $y=\left\{y_{1},y_{2},y_{3},\ldots ,y_{i}\right\}$ where i={1,2,3,…,N}. The IoT sensor datasets with the maximum range “N” are processed in the input layer and transmitted to filter layer to reduce the noise by analysing number of input vectors. In the datasets, there are a number of depolarized and repolarized patterns, which are processed using mean and standard deviation values as shown in the Equation (1),
(1)$$N_{i}=\frac{y_{i}-\bar{y_{i}}}{\sigma }\;,f\left(N_{i}\right)=\{\begin{array}{c} N_{i},\;i<N\\ N_{i},\;i\geq 0 \end{array}$$

From Equation (1), $N_{i}$ is the normalized set of input vectors used in the deep learning network and f(x) is considered as various range starts from zero to “N”. This condition helps to extract the physiological signals, for instance typical electro cardiogram (ECG) tracing, as shown in Figure 4, consists of atrial depolarization wave as denoted as “P” and ventricular depolarization as denoted as “QRS” with ventricular repolarization “T”. In this wave, “U” is generally ignored because it is not typically seen, which represents the papillary muscles patterns. These four entities are analysed from input to output layer and it has been processed using Bayesian network for accuracy.

From the analysis, the means and standard deviations are represented in terms of zero and high logic in Equations (2) and (3),
(2)$${\bar{y}}_{i}=\frac{\sum_{N}^{i}y_{i}}{N}\;f\left(N_{i}\right)=\{\begin{array}{c} N_{i},\;i<N\\ N_{i},\;i\geq 0 \end{array},$$
(3)$$\sigma =\sqrt{\frac{y_{i}-\bar{y_{i}}}{N-1}}.$$

Equations (2) and (3) help to analyse the data description and normalized range of all the health datasets from input layer to output layer. Then, the IoT datasets are processed on the edge platform inside the smart log interface, as discussed in the Case-2.


**Case: 2-IoT data acquisition sensor for smart log system routing mode on edge platform**



**Solution:**


IoT data acquisition sensor architecture consists of a multiplexer, a buffer, and a static random access memory (SRAM) configuration cell, as shown in Figure 5.

As shown in Figure 1, IoT programmable routing sensor architecture consists of n-metal oxide semiconductor (n-MOS) (MN1 and MN2) and p-metal oxide semiconductor (p-MOS) (MP1, MP2, MP3, MP4, and MP5) sleep transistors in parallel. Here, the data selector, so-called multiplexer, has been used to regulate the output data, and data acquisition sensors are used as data-loggers, which maintain the database of the system. In this hardware structure, Both P-MOS and N-MOS operates on three mode of operation such as Dynamic, Sleep and Snooze mode as listed in the Table 1. Here, the IoT data have been analysed using statistical and information gain ratio F(IG) using the health datasets. The information gain is shown in the Equation (4)
(4)$$F\left(IG\right)=IG_{input\_Node}-\overset{N_{I}}{\underset{i=1}{\sum }}\frac{NI\left(y_{i}\right)}{NI}\;F\left(IG_{i}\right)$$
where $\sum_{i=1}^{N_{I}}\frac{NI\left(y_{i}\right)}{NI}$ is the number of instance “NI” with the range of i vary from 1 to $N_{I}.$

Further, the data integrity of the IoT sensor in the hidden layer are measured using entropy values based on various iteration processes, with filter layer helping to reduce the noise and mean residual error. The prediction reliability of the sensor data has been evaluated using information gain ratio, and further improvement in the information ratio can be mathematically evaluated using gain ratio (GR), which is the ratio of F(IG) and its log factor logh(i) as shown in Equation (5),
(5)$$GR=\frac{F\left(IG\right)}{\sum_{i=1}^{N_{I}}h\left(i\right)logh\left(i\right)}$$
where h(i) is the fraction of datasets that are processed in the hidden layer or filtered layer, and I ranges from i={1,2,3,…,N}.

Here, degree of system routing mode-I operational logics are formulated with the help of AND (&) operator and the degrees are shown in Equations (6) and (7),
(6)$$H{\left(L\right)}_{N}=\mu_{1}\left(y_{1}\right)\forall \;\mu_{2}\left(y_{2}\right)\forall \ldots \ldots \mu_{3}\left(y_{3}\right)\forall \;\mu_{N}\left(y_{i}\right)$$
(7)$$H{\left(L\right)}_{N}=\mu_{1}\left(y_{1}\right)*\;\mu_{2}\left(y_{2}\right)*\ldots \ldots \mu_{3}\left(y_{3}\right)*\mu_{N}\left(y_{i}\right)$$
where ∀ is the min operation in the logic. Various modes of operation are listed in the Table 2.

The data retention nature of the switch in the IoT architecture is applied over and done with the tri-modal switch, which has the capacity to preserve information in the snooze mode. The tri-modal switch is intended based on distinct low power IoT design methodologies such as data-retentive power gating, multi-snooze mode structure, and on-chip dynamic voltage scaling as shown in Algorithm 1.


**Algorithm 1.**
*Deep learning-assisted IoT System Routing Mode-I Operation for data processing*

*Initialize inputs MPX, MNX;*

*\ * the MPX, MNX indicated the number of PMOS and NMOS used in the design*\*

*Output S, D, SL;*

*\* The S (Snooze), D (Dynamic), SL (Sleep) mode of operation used*\*
    *Begin*
    *Set1: If (S=Logic ‘0’)*      *MP1|MP3|MP5 =Logic ‘1’;*
      *Else*
      *MN1|MP4 =Logic ‘1’;*
    *Set2: If (S|SL=Logic ‘0’)*      *MN1|MP1|MP3 =Logic ‘1’;*      *Else*      *MP1|MP3|MP5 =Logic ‘1’;*      *Else*      *MN2|MP4 =Logic ‘1’;*    *Set3: If (S|SL=Logic ‘0’)*      *MP2|MP3|MN3 =Logic ‘1’;*      *Else if(S= Logic ‘1’ &&SL=Logic ‘0’)*      *MP2|MP3|MP5 =Logic ‘1’;*      *Else*      *Switch (Set-1);*      *\* Snooze mode due low swing which occurs at Vdd*\*    *End*

As depicted in Algorithm 1 above, the three distinct modes of operation for IoT datasets are shown in Table 2. When sleep mode=Logic ‘1′, irrespective of snooze state, the circuit would be in active state. When snooze mode=Logic ‘0′, it makes the transistor MP1, MP3, MP5 = ON, over which the circuit will attain power through MP4.


**Condition: 1-**


**Data Logic Function: 1-**When Snooze mode=Logic ‘1’, it creates the transistor MN1, MP4 =ON, through which the circuit will attain power through Mp4.


**Condition: 2-**


**Data Logic Function: 2-**When Snooze = Logic ‘0’ and Sleep = Logic ‘0’, it creates the transistor MN2, MP1, MP3 = ON; through this mode, the transistors are activated, which put the circuit in sleep state due to the activation of transistor MN2 = ON, which in turn, stop the passage supply voltage.


**Condition: 3-**


**Data Logic Function: 3-**When drowsy = Logic ‘1′ and sleep = Logic ‘0′MP3, MP2, MN2 = ON because of this method, it creates the circuit worked at drowsy mode due to low swing, which ensues at Vdd1.These modes of operation aid in the decrease in the energy consumption of the storage unit with less delay in the wearable architecture of the smart log patch. The energy consumption has been significantly reduced by reducing the unwanted switching activity of the transistor using drowsy, sleep, and snooze state transitions.

Further, the network training has been done using a Bayesian deep learning algorithm on an edge computing platform processing health datasets to improve accuracy and efficiency as discussed in case 3.


**Case: 3- Training the network using Bayesian deep learning algorithm with prediction metrics**



**Solution:**


Here, the agile learning has been introduced because it provides the trade-off between the data, which have been processed from the input to output layer in terms of complexity or congestion and accuracy. This is because, in general, deep learning models are more complex on edge devices in real-time diagnosis of health data of the multi access physical monitoring system with multimedia technology. Here, the data have been normalized to avoid data distribution and congestion as represented in Equations (8) and (9),
(8)$$K^{th}=y^{k}-F{\bar{y}}^{k}$$
(9)$$D_{N}=\frac{F\left(K^{th}\right)}{\sqrt{Var\left(y^{k}-F{\bar{y}}^{k}\right)}}$$
where $y^{k}-F{\bar{y}}^{k}$ is the difference between dimensions of *k*th input with respect to the average number of input datasets. $\frac{F\left(K^{th}\right)}{\sqrt{Var\left(y^{k}-F{\bar{y}}^{k}\right)}}$ is the ration of *k*th dimension datasets with standard deviation. This is mainly described to reduce the layer of unwanted noise due to external frequency on the smart log system. The computation complexity of the smart log patch has been reduced using agile learning liner activation function, which is considered as one of the significant components in the Bayesian network, as shown in Equations (10) and (11).
(10)$$x^{k}=\alpha^{k}y^{k}-\beta^{k}¥$$
(11)$$\alpha^{k}y^{k}=N_{R1}$$
(12)$$\beta^{k}¥=N_{R2}$$

Substitute (11) and (12) in (10) and we will get a noise-free expression model ($x^{k}$), which minimizes the problem in data complexity using a certain degree of fast convergence, which provides the optimum trade-off between accuracy and complexity as well as reducing the delay, as shown in Equation (13)
(13)$$x^{k}=N_{R1}-N_{R2}$$
where $N_{R1}-N_{R2}$ is the noise factor, which is the product of $\alpha^{k}y^{k}\;\&\;\;\beta^{k}¥$, and where $\alpha^{k}y^{k}\;\&\;\;\beta^{k}¥$ are the input vectors of the linear activation function of agile learning. In this activation function, Gaussian factor has been introduced from input layer to output layer to improve the accuracy of the prediction and helps to reduce the energy consumption by maintaining unwanted switching activities in the network during data processing of the smart log patch. The Gaussian restricted activation function is represented as Equations (14) and (15),
(14)$$F\left(g,h|\emptyset \right)=\overset{N}{\underset{i=1}{\sum }}\frac{v_{n}-h_{n}}{Var\left(\sigma_{i}^{2}\right)}-\overset{k}{\underset{i=1}{\sum }}\frac{W_{n}*h_{n}}{\sigma_{i}^{2}}*\overset{N}{\underset{i=1}{\sum }}\frac{v_{n}}{\frac{1}{\sigma_{i}^{3}}}-\overset{N}{\underset{1=1}{\sum }}\frac{v_{n}}{\sigma_{i}^{2}}$$
(15)$$F\left(g,h|\emptyset \right)=\overset{N}{\underset{i=1}{\sum }}\frac{v_{n}-h_{n}}{Var\left(\sigma_{i}^{2}\right)}-\overset{k}{\underset{i=1}{\sum }}\frac{W_{n}*h_{n}}{\sigma_{i}}v_{n}-\overset{N}{\underset{1=1}{\sum }}\frac{v_{n}}{\sigma_{i}^{2}}$$
where
F(g,h|∅) is the Gaussian restricted activation function;$v_{n}$—visible neurons;$h_{n}$—hidden neurons;$\sigma_{i}$—standard deviation of the Gaussian restricted activation function;$W_{n}$—weight of the neuron.

In agile learning of Bayesian networks, the complexity of the data with time factor as mentioned in n/m seconds is computed using a Bayesian deep learning prediction algorithm. Here, “y” is the input dataset with the total data length “N” and the time is measured as “T” to predict the input data “y”; the complexity can be reduced if we set the time limit “T≤n/m” and the computation flow has been shown in the Algorithm 2.

**Algorithm 2.** Agile learning of Bayesian networks for congestion check in wearable system
*Initial: Time T= ($T_{1},T_{2},T_{3},\ldots ,T_{i}$)*

*Ensure: No congestion on Prediction data for*
$y=\left\{y_{1},y_{2},y_{3},\ldots ,y_{i}\right\}$

*While (Logic “1”) for prediction check*
    *If (j<n) then*
    *M (O) =S (D);*
*//*M (O) = is the memory output layer//**

*//*S (D) = Input datasets which are stored//**
    *S (D) =$\;y=\left\{y_{1},y_{2},y_{3},\ldots ,y_{i}\right\}$*
*Return (No_Congestion)*
    *If (T≤n/m)*    *M (O) = D(C)*
*//*D(C) = Data Congestion//**
      *Return (Congestion)*      *Break*  *M (O) = Return (prediction check)*
*End if*
    *End if*    *  End While*        *End begin*

From Algorithm 2, after the completion of the prediction phase, the final training stage has been formulated for fine tuning the datasets at the output layer. Here, activation function of the neurons are represented in Equation (16),
(16)$$\begin{array}{c} y_{j}=f(w_{1,\ldots ,}w^{N}+\sum_{i=1}^{N}W_{ij}^{N}*Y_{i})\\ i=1,2,3,\ldots ,l\\ j=1,2,3,\ldots ,k \end{array}$$
(17)$$\begin{array}{c} Z_{j}=f(w_{2,\ldots ,}w^{N}+\sum_{i=1}^{N}W_{kj}^{N}*y_{j})\\ k=1,2,3,\ldots ,N \end{array}$$
where,
$w_{1,\ldots ,}w^{N}\to \mathrm{Weight}\;\mathrm{of}\;\mathrm{the}\;\mathrm{neurons}$;l,k,N→are the number of elements in input,hidden and output layer;$Z_{j}=\mathrm{Output}\;\mathrm{layer}\;\mathrm{prediction}$;$x=\left\{x_{1},x_{2},x_{3},\ldots ,x_{i}\right\},y=\left\{y_{1},y_{2},y_{3},\ldots ,y_{i}\right\}\mathrm{are}\;\mathrm{the}\;\mathrm{input}\;\mathrm{vectors}$.

This output factor has been designed to represent the accurate class activities of the neurons with the help of visible nodes in the network on edge platform of the smart log patch.

From the all the cases, it is clear that a smart log patch with a Bayesian deep learning network on an edge computing platform shows promising outcome in terms of accuracy, efficiency, mean residual error, delay, and energy consumption.

Then, the efficiency of this wearable IoT smart log system with multimedia technology is evaluated using experimental results and discussions as follows.

## 4. Experimental Analysis

In this research, various health datasets have been compared by placing a wearable smart log patch, which analyses various activities of the complete body nerves and helps to monitor blood, temperature, electro-myo-gram (EMG), electro-cardio-gram (ECG), electro-encephalo-gram (EEG), pressure, visual, respiration, and accelerator gyroscope of the human physical system through palm and heel because a completed nerve system has been integrated in the palm and leg as listed in Table 3 and the hardware specification of the smart log patch has been given in Table 3.

The hardware details of the wearable IoT smart log patch has been tabulated in Table 4. The datasets are collected from the IoT sensor on the edge platform and it has been analysed to check the performance parameters, which are listed below.

The accurate classification has been done with the help of the activation function. α^k y^k & $\beta^{k}¥$ are the input vectors of the linear activation function of agile learning. In this activation function, Gaussian factor has been introduced from input layer to output layer to improve the accuracy of the prediction and it helps to reduce the energy consumption by maintaining unwanted switching activities in the network during data processing of the smart log patch. The comparison of EC-BDLN with CCT, RNA, CNA, DIA, and CPM shows prominent results in the smart log patch in the output layer, as shown in Figure 6. The accuracy has been calculated based on the true positive and negative values in correlation with false positive and negative values of the sensor datasets, which is represented as Equation (18).
(18)$$Accuracy=\frac{F\left(Tp\right)+F\left(Tn\right)}{F\left(Tp\right)+F\left(Tn\right)+F\left(Fp\right)+F\left(Fn\right)}$$

In the IoT sensor, the error has been estimated as the difference between observed and unobserved quantity of health datasets. Here, mean residual error is the estimation of observed health dataset ($O_{values}$) or processed datasets with the total number of estimated inputs ($P_{values}$). In this research, these layers are integrated with a Bayesian network where the matrix consists of various IoT datasets, which are represented as numerical in the matrix and each layer is connected between the same and cross layer, which are represented as a regression model. Here, the normalization operation has been mathematically derived using “ Mean (µ) and Standard Deviation (σ)”; to reduce the error rate through filtered output, the hidden layer has been processed using gain ratio(GR) as shown in Equation (5), the graphical representation is shown in Figure 7, and the mathematical formulation of RMSE is shown in Equation (19),
(19)$$RMSE=O_{values}-P_{values}$$
(20)$$PSNR=10log_{10}\left(\frac{R^{2}}{Residual\;Mean\;Square\;Error}\right).$$

As shown in Equation (20), the mean residual square error (RMSE) is the collective squared error among the input and output image, whereas peak signal-to-noise ratio (PSNR) indicates a degree of the peak error. The lower the value of RMSE, the lower the error. These, in turn, increase the efficiency of the system. Because the data retention of the switch in the IoT architecture is applied over the tri-modal switch, which has lowers RMSE, the graphical comparison of efficiency of the proposed system is shown in Figure 8.

The number of health dataset inputs and its corresponding rate of transmission as estimated as delay in the IoT data transmission delay have been reduced due to the following conditions, which are listed as:


***Delay factor***
C1- Snooze mode = Logic ‘1′ it makes the transistor MN1, MP4 = ON;C2-Snooze = Logic ‘0′ and Sleep = Logic ‘0′ it makes the transistor MN2, MP1, MP3 = ON;C3- drowsy = Logic ‘1′ and sleep = Logic ‘0′MP3, MP2, MN2 = ON.


The comparison of EC-BDLN with CCT, RNA, CNA, DIA, and CPM shows prominent results in the smart log patch in the output layer as shown in Figure 9.

In this graph, as shown in Figure 10, the proposed EC-BDLN algorithm reduces data faults and also has high throughput while sending the information from input to output layer. Even though the network effectively detects data faults, it successfully transmits the data from source to destination by consuming minimum energy, where the Gaussian factor has been introduced to improve the accuracy of prediction by maintaining unwanted switching activities in the network during data processing of the smart log patch. The Gaussian restricted activation function is represented as Equations (14) and (15) and the energy table is shown in Table 5.

From the results and discussions, it shows that the EC-BDLN algorithm is one of the state-of-the-art evolutionary algorithms in health monitoring of multi access physical monitoring system with multimedia technology. In this research, a wearable smart-log patch with Internet of Things (IoT) sensors has been designed and developed with multimedia technology to analyse the various activities of complete body such as blood, temperature, electro-myo-gram (EMG), electro-cardio-gram (ECG), electro-encephalo-gram (EEG), pressure, visual, respiration, and accelerator gyroscope of the human physical system, and the optimization parameters such as accuracy, efficiency, mean residual error, delay, and energy consumption have been experimentally validated using the EC-BDLN algorithm in the distributed devices on an edge computing environment, which shows to be more promising than traditional approaches.

## 5. Conclusions

Presently, information requirements in multi access physical monitoring system and its effective diagnosis of health condition is the challenging task in practice. In this article, a wearable smart-log patch with Internet of Things (IoT) sensors has been designed and developed with multimedia technology to overcome the setback faced by the current methods in multi access physical monitoring systems. In this research, a complete physical monitoring system using multimedia technology with edge computing using agile learning for real-time data analysis using IoT sensors has been hybridized with a Bayesian network for accurate prediction of improper working of organs, which are integrated in a wearable IoT smart patch for data processing. The progress of edge computing on a Bayesian neural network (EC-BNN), which helps to infer and identify various physical data collected from humans with high prediction in classifying health datasets or physical activity information, is deployed as one of the promising choices to solve the multi access physical monitoring system problem. In future, intelligent Internet of Things are planned to be integrated for further up scaling of the device, with advanced multimedia techniques to reduce the cost factor and privacy.

## Figures and Tables

**Figure 1 sensors-19-03030-f001:**
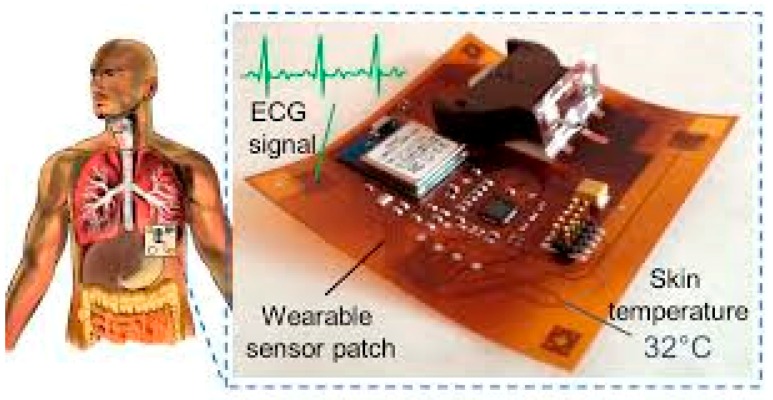
Ultra-thin flexible patch array for physical monitoring.

**Figure 2 sensors-19-03030-f002:**
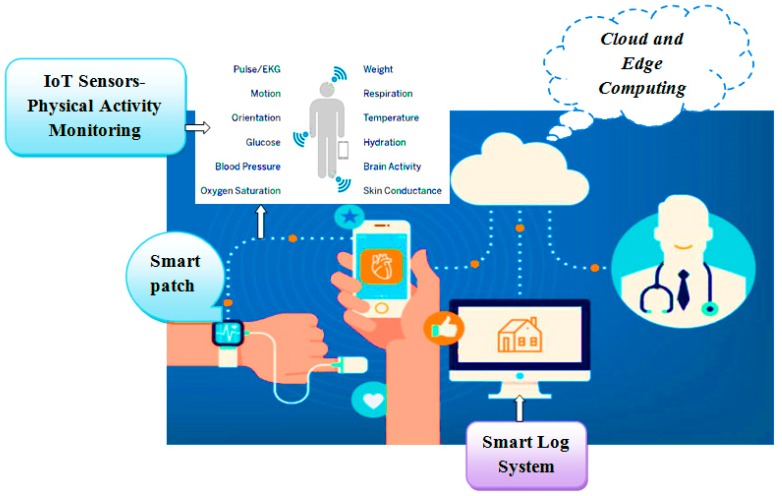
Wearable smart log patch with Internet of Things (IoT) sensor in edge computing environment.

**Figure 3 sensors-19-03030-f003:**
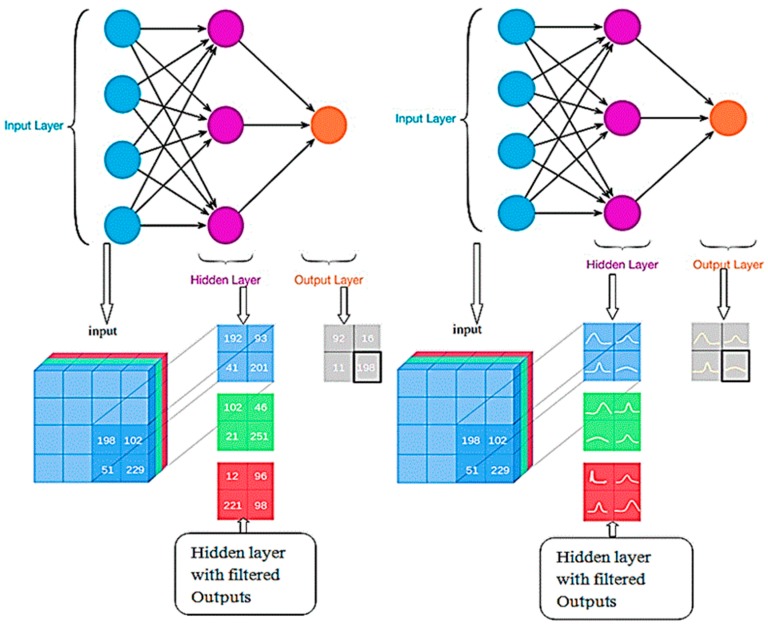
Bayesian deep learning structural block.

**Figure 4 sensors-19-03030-f004:**
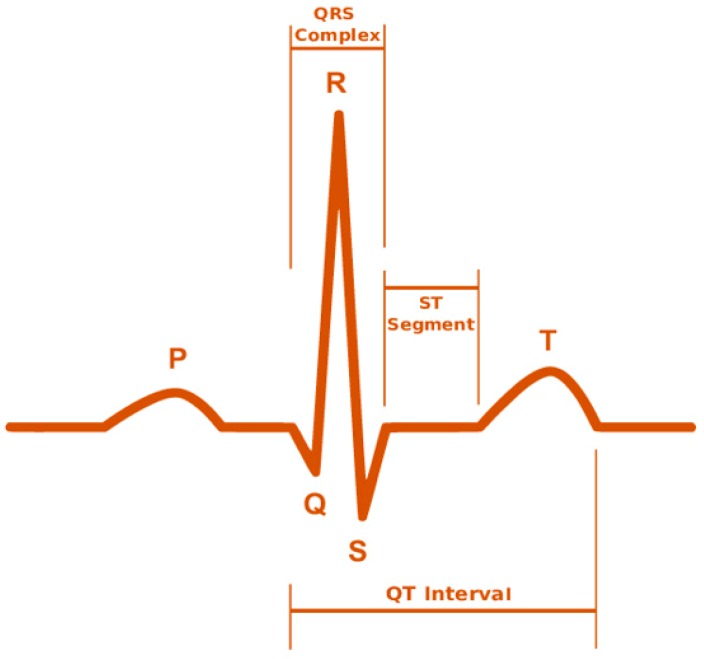
Electro cardiogram (ECG) tracing.

**Figure 5 sensors-19-03030-f005:**
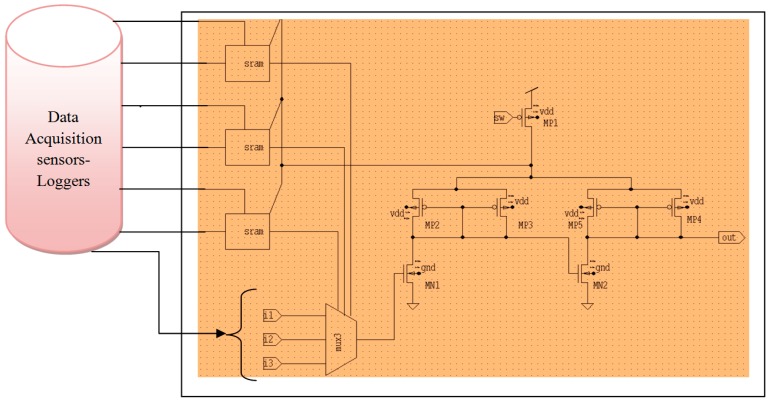
IoT data acquisition sensor architecture.

**Figure 6 sensors-19-03030-f006:**
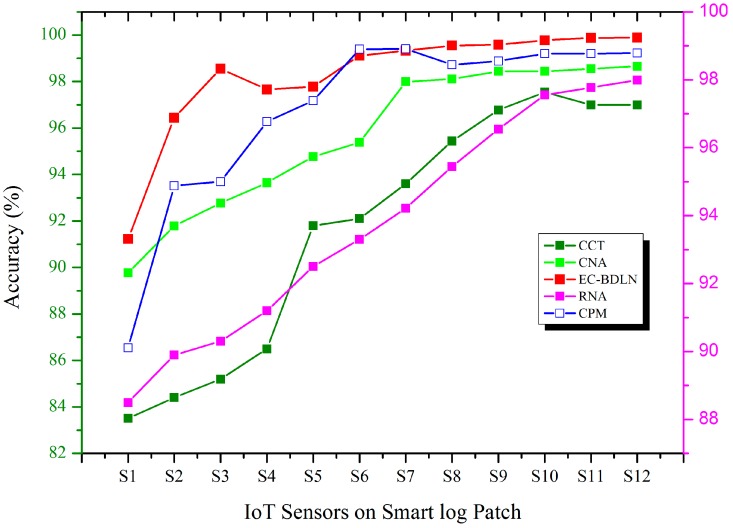
Accuracy factor of IoT sensor.

**Figure 7 sensors-19-03030-f007:**
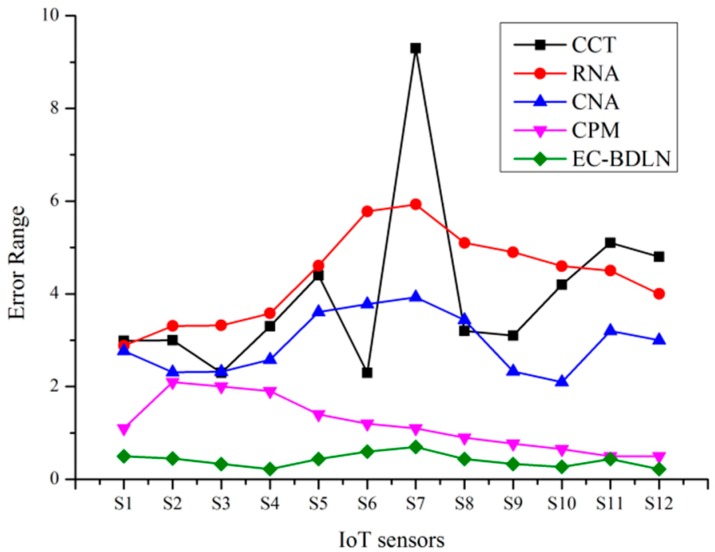
Mean residual error estimation analysis for the smart log patch.

**Figure 8 sensors-19-03030-f008:**
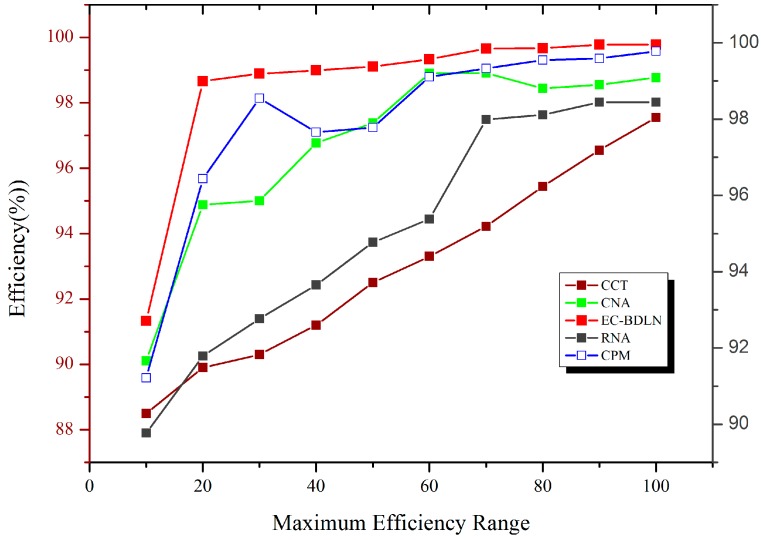
Estimation using mean residual square error analysis.

**Figure 9 sensors-19-03030-f009:**
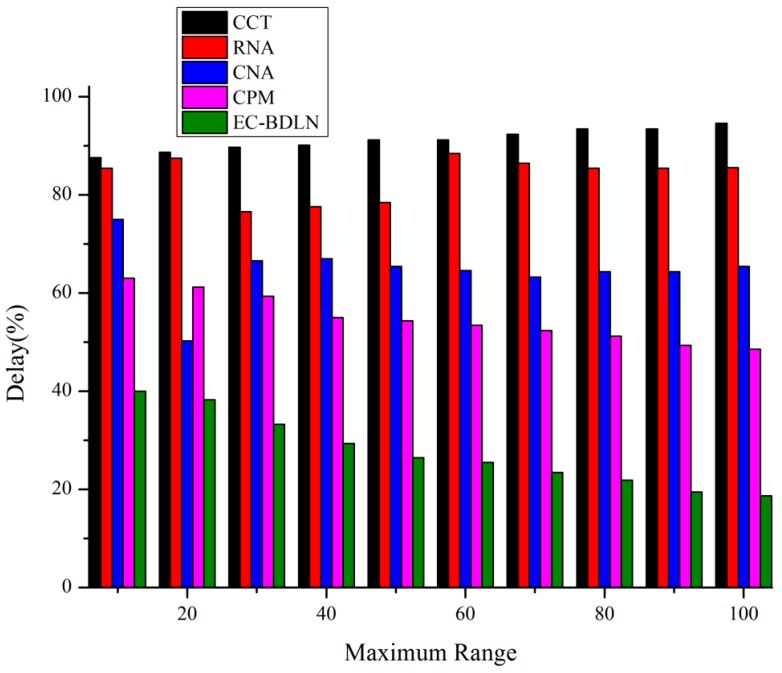
IoT data transmission delay factor of edge computing on Bayesian deep learning network (EC-BDLN).

**Figure 10 sensors-19-03030-f010:**
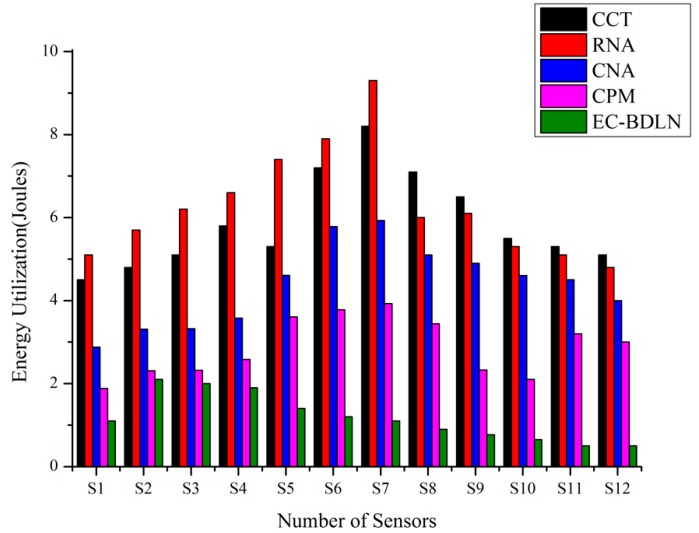
Energy utilization factor of EC-BDLN.

**Table 1 sensors-19-03030-t001:** IoT sensor signal estimation.

IoT Sensor	Usage	Representation of Datasets
Blood Pressure	Through Photo-plethysmo-graph (PPG) pressure and temperature of the body has been monitored. PPG is the optical technique that analyses the micro vascular bed of the tissue.	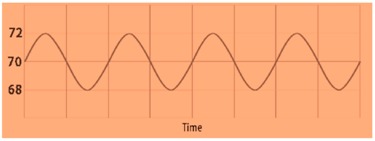
Body Temperature		
ECG	Electrical activity of heart over the period of time includes contraction and relaxation	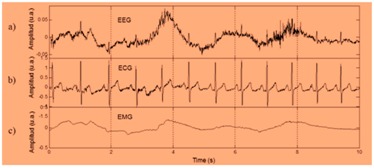
EEG	To check the brain activity of the person	
EMG	Electrical activity of the muscles	
Pressure	Based on pulse transit time, the body pressure is monitored	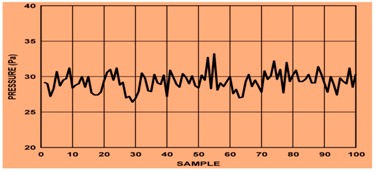
Visual	To check the interpretation of objects and data	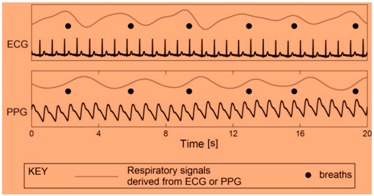
Respiration	To check the breathing patters of a person	
Accelerator gyroscope	To analyse the inclination of the body	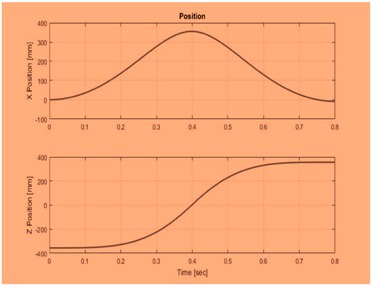

**Table 2 sensors-19-03030-t002:** Deep learning-assisted system routing mode-I operation.

System Routing Mode-I		
Sleep	**Snooze**	**Function**
Logic-′1′	Undetermined	Dynamic
Logic-′0′	Logic-′0′	Sleep
Logic-′0′	Logic-′1′	Snooze

**Table 3 sensors-19-03030-t003:** Wearable smart log patch on the wrist—signal analysis.

Symbol of Sensors	IoT Sensor Signal Check on The Leg and Hand
S1	Analyse the Left lower palm
S2	Analyse the Left upper palm
S3	Analyse the Right upper palm
S4	Analyse the Right lower palm
S5	Analyse the Left lower heel
S6	Analyse the Left upper heel
S7	Analyse the Right upper heel
S8	Analyse the Right lower heel
S9	Centre of the backside below spinal cord
S10, S11, S12	Calf region (Right and Left)

**Table 4 sensors-19-03030-t004:** Hardware specification of the wearable IoT smart log patch.

Characteristic	Model-1 (as Shown in the Figure 1)
**Voltage**	0.9 V
**Chip dimension**	5 mm
**Clock Speed**	120 Mhz
**Build in Wi-Fi and Bluetooth**	Yes
**Digital I/O Pins**	14 Numbers
**Number of sensors**	12 Numbers

**Table 5 sensors-19-03030-t005:** Energy utilization of various algorithms.

	Energy Utilization (Joule)				
Number of Sensors	CCT	RNA	CNA	CPM	EC-BDLN
**S1**	4.5	5.1	2.88	1.88	1.1
**S2**	4.8	5.7	3.31	2.31	2.1
**S3**	5.1	6.2	3.32	2.32	2.0
**S4**	5.8	6.6	3.58	2.58	1.9
**S5**	5.3	7.4	4.61	3.61	1.4
**S6**	7.2	7.9	5.78	3.78	1.2
**S7**	8.2	9.3	5.93	3.93	1.1
**S8**	7.1	6.0	5.1	3.44	0.9
**S9**	6.5	6.1	4.9	2.33	0.77
**S10**	5.5	5.3	4.6	2.1	0.65
**S11**	5.3	5.1	4.5	3.2	0.5
**S12**	5.1	4.8	4.0	3.0	0.5

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
