# Peer review of "Wearable IoT Smart-Log Patch: An Edge Computing-Based Bayesian Deep Learning Network System for Multi Access Physical Monitoring System"

_sensors, 2019, doi:10.3390/s19133030_

Round 1
Reviewer 1 Report
The paper presents a wearable patch featuring IoT facilities. The data are analyzed using Edge computing on Bayesian deep learning network to infer and identify various physical data collected from humans.
In the "introduction" and in "Related Works" the energy consumption problem is highlighted. This topic should be more detailed. There are works that present possible solutions. For example, the following papers are related:
- Yang, G., Jiang, M., Ouyang, W., Ji, G., Xie, H., Rahmani, A.M., Liljeberg, P., Tenhunen, H.
IoT- Based Remote Pain Monitoring System: From Device to Cloud Platform (2018) IEEE Journal of Biomedical and Health Informatics, 22 (6), art. no. 8118086, pp. 1711-1719.
- Bassoli, M., Bianchi, V., De Munari, I. A plug and play IoT Wi-Fi smart home system for human monitoring (2018) Electronics (Switzerland), 7 (9), art. no. 200
The section "A. Contribution" should be moved in the "Introduction" section as a text describing the goal of the paper.
A more detailed description of the hardware is strongly suggested in a dedicated section.
A more detailed description of the measurements performed to obtain the energy consumption data must be reported.
The paper should be carefully reread by the authors, a lot of grammatical and typo errors are present that make the reading difficult.
- Please define all the acronyms, PES is not defined.
- don't use capital letter after Figure xx or eq(xx)
- rephrase the sentence from lines 62-65. It is not clear
- line 94: Experimental -> experimental
- line 116: Conditional based monitoring (CBM) -> Conditional Based Monitoring (CBM)
- line 174: the sentence is truncated, it must be merged with the beginning of line 191.
- line 196: represent -> represented
- line 253: rephrase the sentence
- line 254 and 256: "As shown in the table 2" is repeated twice.
- line 393 and line 395: rephrase the sentences.
- line 400: correct the sentence, the function is not readable.
- line 406: tha accuracy been calculated -> the accuracy has been calculated
- line 429: are represent -> are represented
- Table 4: unit of measurement is not reported
Author Response
The paper presents a wearable patch featuring IoT facilities. The data are analyzed using Edge computing on Bayesian deep learning network to infer and identify various physical data collected from humans.
In the "introduction" and in "Related Works" the energy consumption problem is highlighted. This topic should be more detailed. There are works that present possible solutions. For example, the following papers are related:
- Yang, G., Jiang, M., Ouyang, W., Ji, G., Xie, H., Rahmani, A.M., Liljeberg, P., Tenhunen, H. IoT-Based Remote Pain Monitoring System: From Device to Cloud Platform (2018) IEEE Journal of Biomedical and Health Informatics, 22 (6), art. no. 8118086, pp. 1711-1719.
- Bassoli, M., Bianchi, V., De Munari, I. A plug and play IoT Wi-Fi smart home system for human monitoring (2018) Electronics (Switzerland), 7 (9), art. no. 200
Ans: The contributions of the above mentioned paper has been included in the survey
The section "A. Contribution" should be moved in the "Introduction" section as a text describing the goal of the paper.
Ans: It has been updated as per the suggestion
A more detailed description of the hardware is strongly suggested in a dedicated section.
Ans: Details has been updated and listed out in the methodology section
A more detailed description of the measurements performed to obtain the energy consumption data must be reported.
Ans: The description has been updated and reported
The paper should be carefully reread by the authors, a lot of grammatical and typo errors are present that make the reading difficult.
Ans: it has been checked and verified
- Please define all the acronyms, PES is not defined.
Ans: PES acronym is updated in the manuscript
- don't use capital letter after Figure xx or eq(xx)
Ans : Changed as per suggestions
- rephrase the sentence from lines 62-65. It is not clear
Ans: rephrased
- line 94: Experimental -> experimental
Ans: checked and changed it as experimental
- line 116: Conditional based monitoring (CBM) -> Conditional Based Monitoring (CBM)
Ans: Checked and updated as Conditional Based Monitoring (CBM)
- line 174: the sentence is truncated, it must be merged with the beginning of line 191.
Ans: It is checked and updated as per suggestions
- line 196: represent -> represented
Ans: It is checked and changed
which are represented as signals and these signals are converted as numerical in the matrix and each layers are connected between same and cross layer which are represented as regression model.
- line 253: rephrase the sentence
Ans: It is checked and rephrased
- line 254 and 256: "As shown in the table 2" is repeated twice.
Ans: Checked and changed it as (Where the min operation in the logic and various modes of operation is has been listed in the Table.2)
- line 393 and line 395: rephrase the sentences.
Ans: Checked and rephrased as per suggestions
- line 400: correct the sentence, the function is not readable.
Ans: Checked and Updated
- line 406: tha accuracy been calculated -> the accuracy has been calculated
Ans: Checked and Updated as (The Accuracy has been calculated based on the true positive and negative values in correlation with false positive and negative values of the sensor data sets which is represented as Eq (19).)
- line 429: are represent -> are represented
Ans: Checked and Updated it is are represented
- Table 4: unit of measurement is not reported
Ans: Energy Utilization (Joule)
Reviewer 2 Report
I suggest accepting manuscript for the publication, after minor revision:
1. State the main goal of the manuscript in the Introduction section.
2. Improve Conclusion section with scientific contribution of the manuscript.
3. Spell check required.
Author Response
1. State the main goal of the manuscript in the Introduction section:
Ans: The major contribution is listed in the introduction part:
A novel optimized neutral network with densely connected layer for determining the temperature imbalance in health.
Bayesian deep learning Network for accurate prediction of improper working of organs which are integrated in Wearable IoT smart patch for data processing.
Complete physical monitoring system using multimedia technology with edge computing using agile learning for real time data analysis using IoT sensors.
Streamlined efficient model to identify the various signal patterns of the human physical activities using Edge computing on Bayesian neural network.
2. Improve Conclusion section with scientific contribution of the manuscript.
Ans: It is improved with scientific method used in this article and listed
In this research complete physical monitoring system using multimedia technology with edge computing using agile learning for real time data analysis using IoT sensors has been hybridized with Bayesian Network for accurate prediction of improper working of organs which are integrated in Wearable IoT smart patch for data processing. The progress of Edge computing on Bayesian neural network (EC-BNN) which helps to infer and identify various physical data collected from the humans with high prediction in classifying health data sets or physical activity information is deployed as one of the promising choice to solve Multi access Physical monitoring system problem.
2. Spell check required.
Ans: checked and Verified.
Round 2
Reviewer 1 Report
I am satisfied with the changes performed by the authors. Therefore, I recommend to accept the manuscript in its present form.